# Effectiveness of healthcare workers and volunteers training on improving tuberculosis case detection: A systematic review and meta-analysis

Desalegne Amare[1]*, Fentie Ambaw Getahun[2], Endalkachew Worku Mengesha[2], Getenet Dessie[1], Melashu Balew Shiferaw[3], Tegenaw Asemamaw Dires[4], Kefyalew Addis Alene[5,6]

1 School of Health Sciences, College of Medicine and Health Sciences, Bahir Dar University, Bahir Dar, Ethiopia, 2 School of Public Health, College of Medicine and Health Sciences, Bahir Dar University, Bahir Dar, Ethiopia, 3 Amhara Public health Institution, Bahir Dar, Ethiopia, 4 Management Sciences for Health, Addis Ababa, Ethiopia, 5 Telethon Kids Institute, Nedlands, Western Australia, Australia, 6 Faculty of Health Sciences, Curtin University, Bentley, Western Australia, Australia

* desalegnezelellw@gmail.com

**Data Availability Statement:** All relevant data are within the paper and its Supporting Information files.

## Abstract

### Introduction

Tuberculosis is the second most common infectious cause of death globally. Low TB case detection remains a major challenge to achieve the global End TB targets. This systematic review and meta-analysis aimed to determine whether training of health professionals and volunteers increase TB case detection.

### Methods

We performed a systematic review and meta-analysis of randomized control trials and non-randomized control trials reporting on the effectiveness of health professionals and volunteers training on TB case detection. We searched PubMed, SCOPUS, Cochrane Library, and reference sections of included articles from inception through to 15 February 2021, for studies published in English. Study screening, data extraction, and bias assessments were performed independently by two reviewers with third and fourth reviewers participating to resolve conflicts. The risk of bias was assessed using the Joanna Briggs Institute (JBI) checklist. Meta-analyses were performed with a random effect model to estimate the effectiveness of training intervention on TB case detection.

### Results

Of the 2015 unique records identified through our search strategies, 2007 records were excluded following the screening, leaving eight studies to be included in the final systematic review and meta-analysis. The results showed that providing training to health professionals and volunteers significantly increased TB case detection (RR: 1.60, 95% CI: 1.53, 1.66).

**Funding:** The authors received no specific funding for this work.

**Competing interests:** The authors have declared that no competing interests exist.

**Abbreviations:** CD, Case detection; JBI, Joanna Briggs Institute; PRISMA, Preferred Reporting Items for Systematic Reviews and Meta-Analyses; MeSH, Medical Subject heading; RCT, randomized control trial; TB, Tuberculosis; WHO, World Health Organization.

There was not a significant degree of heterogeneity across the included study on the outcome of interest ($I^2$ = 0.00%, p = 0.667).

## Conclusions

Providing training to healthcare workers and volunteers can increase TB case detection.

## Introduction

Despite Tuberculosis (TB) is a preventable and treatable disease, it remains an important cause of death from an infectious agent [1], especially, TB, the second-deadliest infectious killer (after COVID-19), is caused by Mycobacterium tuberculosis, which primarily affects the lungs [2]. Based on the World Health Organization (WHO) report, approximately 10 million people who developed TB in 2020. More than 4 million people have tuberculosis but have not been diagnosed or have not reported the disease to national authorities, which means 40% of all incident cases were not reported to the national tuberculosis program [2,3]. India (41%), Indonesia (14%), the Philippines (12%), and China (8%) were the countries that contributed the most to the global reduction in TB notifications between 2019 and 2020 [2].

Failure of TB case detection can increase the risk of death, severe illness, and transmission of TB in households and communities [4–7]. Previous studies showed that missed pulmonary TB cases can transmit the infection to 10–15 people per year, a major challenge to achieving the global End TB targets [8–10].

The reason for low TB case detection can be related to the healthcare system or patient-related factors. Lack of f adequate training of health professionals on the diagnosis and treatment of TB a common healthcare system factor affecting TB case detection [11–14]. Different interventions were designed to enhance TB case detection in the national TB programs. Most of the interventions designed to improve TB case detection were targeted to TB patients and the communities [15,16]. Extensive advocacy and awareness-raising activities have been implemented to educate people about TB diagnosis [1]. Patient education can be as simple as a booklet or as comprehensive as multiple session programs. However, this approach alone does not increase TB case detection. Additional measures, such as training of health professionals, health extension workers, and community volunteers might be required to increase TB case detection. While few studies have investigated the effects of providing training to health professionals and volunteers on TB case detection, the findings were inconsistent across studies [17–19]. To our knowledge, no systematic review has examined whether health professionals and volunteers training is an effective intervention to increase TB case detection. Therefore, this systematic review and meta-analysis aimed to determine whether training of health professional and volunteers increase TB case detection.

## Materials and methods

### Eligibility criteria

Studies were included if they were randomised controlled trials; non-randomised trials with at least a defined intervention and parallel control groups; quasi-experimental studies; controlled before-after studies with outcome measures before and after the intervention. The interventions were a provision of training to health professionals or any volunteers for at least three days. Potential comparators were usual care, or no intervention (i.e. no training). We excluded conference and meeting abstracts, and non-English language articles, animal studies and those

**Table 1. The characteristics of the studies included in the systematic review and meta-analysis.**

| Study | Study design | Country | Target population | Diagnostic approach | Total sample | Intervention group | | Control group | | Outcomes |
|---|---|---|---|---|---|---|---|---|---|---|
| | | | | | | PIG | CIG | PCG | CCG | |
| Fairall LR et al. 2005 [27] | Cluster RCT | South Africa | Adults | Sputum smear microscopy two times | 1782 | 892 | 57 | 890 | 34 | Sputum screening for tuberculosis was higher among. Patients in the intervention arm but not significantly. During the three months of the study period, 57 new cases of tuberculosis were diagnosed in intervention clinics compared with 34 in control clinics. |
| Ayles et al. 2013[28] | Cluster RCT | South Africa | Adults | Sputum smear microscopy and culture | 962655 | 447228 | 1730 | 515427 | 236 | The prevalence of culture-confirmed pulmonary tuberculosis in adults was increased. |
| Datiko DG and Lindtjørn B. 2009 [18] | Cluster RCT | Ethiopia | All ages | Sputum smear microscopy | 296811 | 178138 | 230 | 118673 | 88 | The mean CDR was higher in intervention kebeles. |
| Talukder K et al. 2012[32] | Cluster RCT | Bangladesh | Child | Keith Edwards TB score was used as a diagnostic test | 3460 | 1577 | 175 | 1883 | 130 | TB notification rates increased significantly more in the intervention (from 3.8% to 12%) than in the control centers. Sputum smear-negative TB was a significant increase in the number of children diagnosed with sputum smear-negative pulmonary TB in the intervention centers. |
| Shargie EB et al. 2006 [30]. | Cluster RCT | Ethiopia | All ages | Sputum smear microscopy | 54405 | 18950 | 24 | 33455 | 33 | All forms of TB notification rate was 125 (159/127 607) in intervention and 98 (221/225 284) in the control groups. |
| Datiko DG et al.2017 [29] | Non-RCT | Ethiopia | All ages | Sputum smear microscopy | 4700000 | 3500000 | 27918 | 1200000 | 5483 | All forms of TB increased from 102 to 177 per 100 000 population in the intervention group. |
| Joshi B et al. 2015 [33] | Non-RCT | Nepal | Child | Sputum smear microscopy, chest X-ray, and tuberculin skin test | 2211909 | 1489425 | 360 | 722484 | 113 | The proportion of all TB cases, the intervention districts showed a significant increase in childhood TB between Years 1 and 2 (from 3.9% to 5.0%), while the control districts showed no significant difference (from 4.2% to 4.9%). |
| Reddy KK et al.2016 | Non-RCT | India | All ages | | 11784 | 7823 | 658 | 3961 | 255 | Smear positive case detection was increased. |
| Oshi DC et al. 2016 [31]. | Non-RCT | Nigeria | Child | Sputum smear microscope, Keith Edwards's child tuberculosis scores. | 25,258,888 | 14742185 | 1067 | 10516703 | 483 | Tuberculosis cases in the intervention areas were increased as compared to the control groups. |

that had insufficient information on the main outcome of interest. The outcome of interest was the TB case detection rate. Our research eligibility in the PICO (Population, Intervention, Comparator, Outcome) format is included in (Table 1).

## Search strategy, information sources and selection criteria

We conducted an electronic medical literature search on PubMed, Cochrane Library, and Scopus databases for relevant randomized controlled trials describing training as an intervention

to increase TB case detection, published from the date of each database inception to 15 February 2021. The review was performed according to the standard procedures of the Cochrane Collaboration [20]. The Preferred Reporting Items for Systematic Reviews and Meta-Analyses (PRISMA) checklist was used to report the review [21]. The protocol was registered in PROS-PERO with registration number CRD42021284106. A comprehensive search strategy was developed using the search terms and medical subject headings [22]. The complete PubMed search strategy is provided in the (S1 File). Reference lists of included papers were screened for additional relevant articles using forward and backward citation search. Corresponding authors were contacted by email in some instances when additional information was required. Key terms were used to search a library database (S1 Table).

## Study selection

All studies identified through the search strategies were imported into EndNote version 7 for screening. After removing duplicate articles, screening of papers by title and abstract were carried out independently by two researchers (DA and GD). Screening of full-text papers was also carried out by the same two researchers for inclusion based on pre-defined eligibility criteria. Disagreement on the inclusion or exclusion of selected papers was resolved through discussion and consensus with co-authors (MBS, EWM, TAD, and FAG).

## Data extraction

Data were extracted from included studies using Microsoft Excel 2016 spreadsheet (Microsoft, Redmond, Washington, USA) by two researchers (DA, GD,). The spreadsheet was checked and refined before extraction began. Data extracted from each study included: name of the first author, year of publication, study design, place of study, type of participant, type of intervention (comparison groups and sample size of each group), outcomes (case detection rate), and the key findings. Information on the duration of implementation, frequency of interventions, site of intervention, and the number of interventions were also extracted.

## Outcome measurement

The primary outcome was TB case detection, expressed by the proportion of all new cases of TB with a confirmed diagnosis.

## Quality assessment

The quality of the included studies was independently assessed by two investigators (DA and EWM) using Joanna Briggs Institute (JBI). The JBI checklist is designed to assess 13 components of RCT study design including randomization method, allocation concealment, baseline characteristics, patient blinding, therapist blinding, observer blinding, co-intervention control, compliance, attrition rate, end-point assessment time point, intention to treat analysis, the same outcome measurement way and reliable outcome measurement [23]. The authors considered low risk of bias 'Yes' either the study protocol is available and all of the studies' pre-specified outcomes that are of interest in the review have been reported in the pre-specified way or the study protocol is not available but it is clear that the published reports include all of the study's pre-specified outcomes and all expected outcomes that are of interest in the review. Authors considered high risk of bias 'No' if any of the following is fulfilled: Not all of the study's pre-specified primary outcomes have been reported, one or more primary outcomes is reported using measurements analysis methods or subsets of the data that were not pre-specified, one or more reported primary outcomes were not pre-specified, one or more outcomes

of interest in the review are reported incompletely, the study report fails to include results for a key outcome that would be expected to have been reported for such a study. Authors considered unclear if there is insufficient information to permit judgment of 'Yes' or 'No'.

Other sources of bias such as recruitment bias were considered as "No" if bias is detected due to problems not covered elsewhere and "Yes" if no other bias is detected. Blinding was classified as "Yes" if steps were taken to ensure that those recording the main outcome of the study were blind to the assigned interventions, and "No" if this was not the case, or if there was no description of the method for assessing the adequacy of the randomization procedure. Completeness of follow-up was assessed as "Yes" if steps to the handling of incomplete outcome data were complete and unlikely to have produced bias or "No" if the attrition amount or handling of incomplete outcome data was not maintained.

The instrument or tool used to assess the risk of bias, rigor, or study quality was reported along with some summary estimate of the quality of primary studies in the included research synthesis [24]. Therefore, the quality of each study was assessed based on JBI quality assessment criteria (S2 Table).

## Data analysis

Meta-analysis with a random effect model was performed to calculate relative risk (RR) with 95% confidence intervals [25]. Heterogeneity between studies was assessed by the index of heterogeneity squared ($I^2$) statistics with 95% CI. The $I^2$ statistic measures the proportion of observed variance between trials that is not due to chance (rather due to real differences across studies populations and interventions). The $I^2$ value less than 30% was interpreted as low evidence of heterogeneity, between 30% and 60% was moderate heterogeneity and $I^2$ more than 60% was interpreted as evidence of substantial heterogeneity [26]. Meta-regression was used to explore whether study characteristics explained heterogeneity. Publication bias was assessed qualitatively by visual inspection of funnel plots and quantitatively by Egger's test. Sensitivity analysis was also conducted based on the quality and characteristics of the studies. The data analysis was performed by Stata version 16. All tests were 2-tailed, and $P < 0.05$ was considered statistically significant.

## Results

### Study selection

Of 4052 studies obtained from the database search, 2037 articles were excluded due to duplications. After removing duplicates, 2015 articles were screened by titles and abstracts and 1946 articles were excluded which resulted in 69 potential articles for full-text review. After a full-text review, 9 studies met the eligibility criteria and were included in the final analysis (Fig 1).

### Study characteristics

Table 1 summarizes the characteristics of the included studies. Studies were conducted in five different countries. All the studies were from African and Asian continents. The population groups were children (under the age of 15) for three studies [29–31], adults for two studies [27,28], and both adults and children for four studies [18,19,29,30]. Interventional trials were included in this analysis. Five of the studies were cluster RCTs, whereas the other four were non–RCTs.

In all studies, the training provided for health workers and volunteers were focused on symptoms, diagnostic and screening methods, and treatment of TB cases. The duration of the

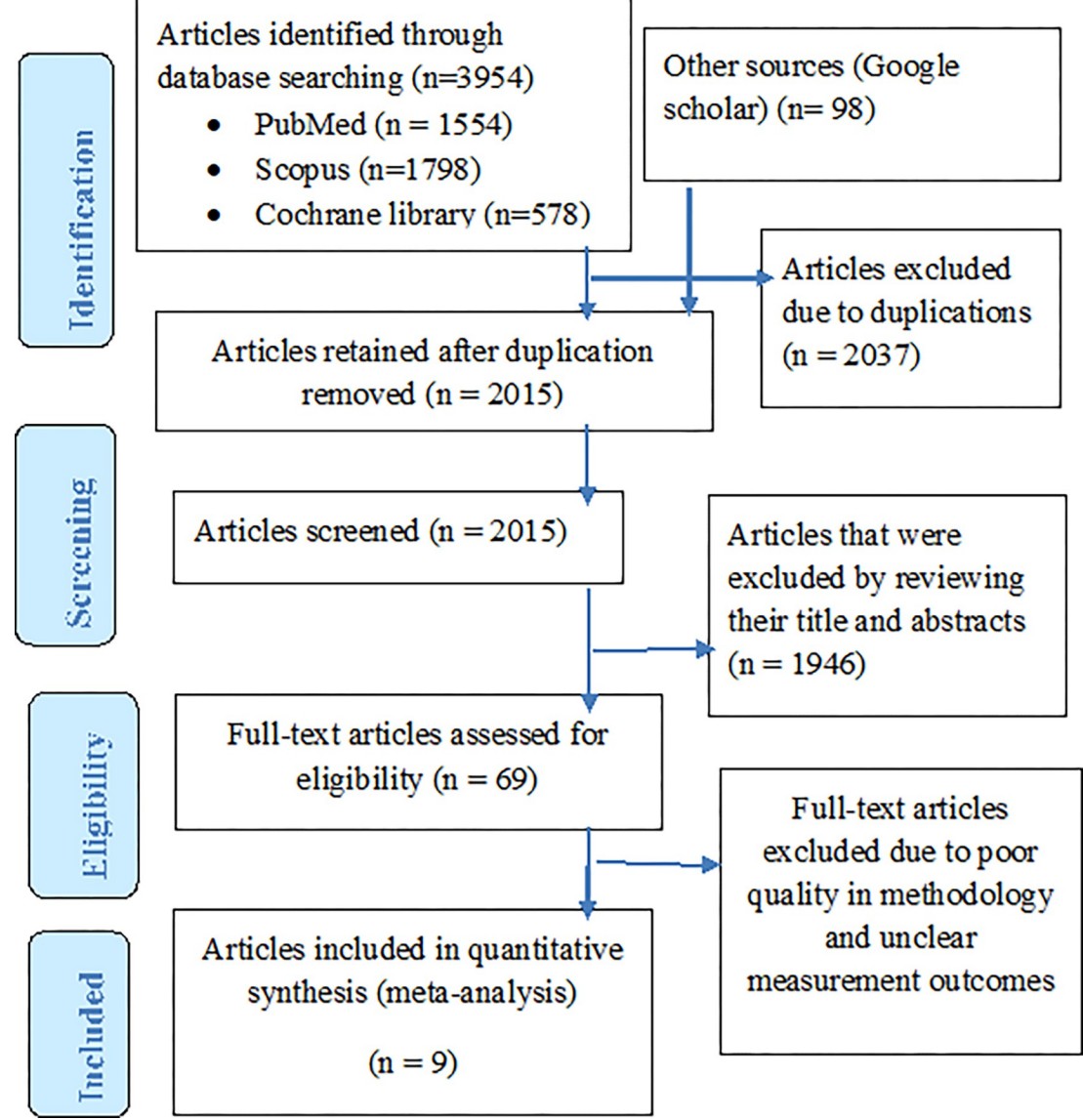

**Fig 1. PRISMA flow chart of study selection.**

intervention varied from 3 months [27] to 54 months [28,29]. Sputum smear microscopic and gene Xpert were used to diagnose TB (Table 1).

Most of the studies included in this study used instructor-led training, demonstration, hands-on training, group-based training, job training, and technology-based training (text message), but the method used by the three studies [19,28,31] is not clear. As a result, it is difficult to determine which training method is more effective in identifying TB case detection.

### Effectiveness of healthcare workers and volunteers training on improving TB case detection

Nine interventional studies were included to investigate whether the intervention group had an effective improvement in TB case detection rate as compared to the control arms. Seven out of nine selected studies were effective to improve the case detection while the other two studies

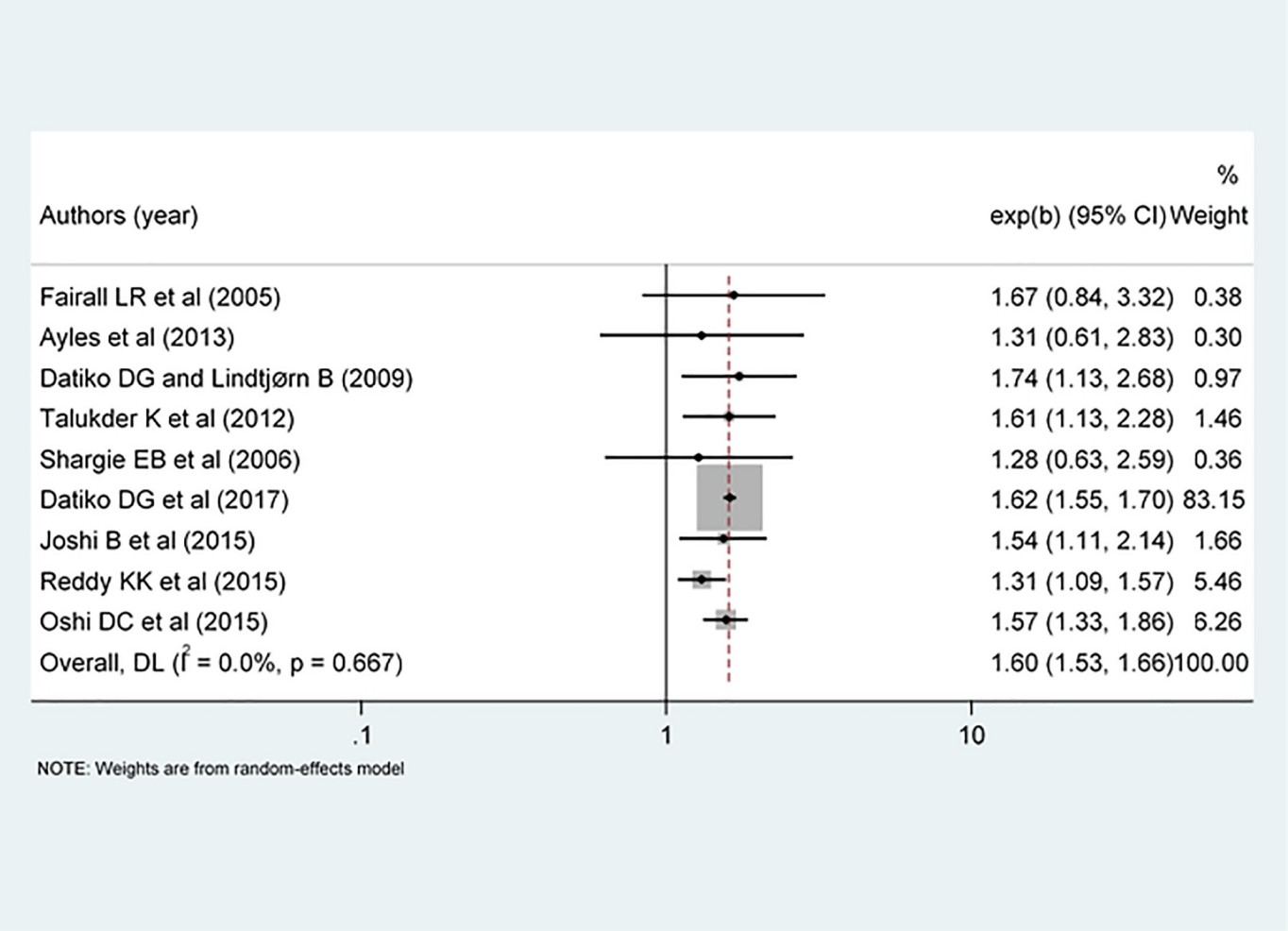

**Fig 2. Forest plot of the effect size of training on tuberculosis case detection in TB high burden settings.**

showed that the interventions had not improved the case detection rate. The overall pooled effect size estimate showed that healthcare workers and volunteers training was significantly improved TB case detection (RR: 1.60, 95% CI: 1.53, 1.66) (Fig 2). In addition, sensitivity analysis showed that there was no single study that affects the overall effect of the intervention (Fig 3).

## Heterogeneity and publication bias

There was no a significant degree of heterogeneity across the included study on the outcome of interest ($I^2$ = 0.00%, p=0.667) (Fig 2). Neither the funnel plot (Fig 4) nor the egger tests showed evidence of significant publication bias (P = 0.244).

## Risks of bias assessment

The risk of bias assessment showed that seven studies had a low risk of bias for random sequence generation and adequate allocation concealment. A blind outcome assessment was reported from only one study (28). In all studies, there was no relevant outcome reporting bias observed. Also, there was no reporting in attrition and selective reporting bias (Table 2).

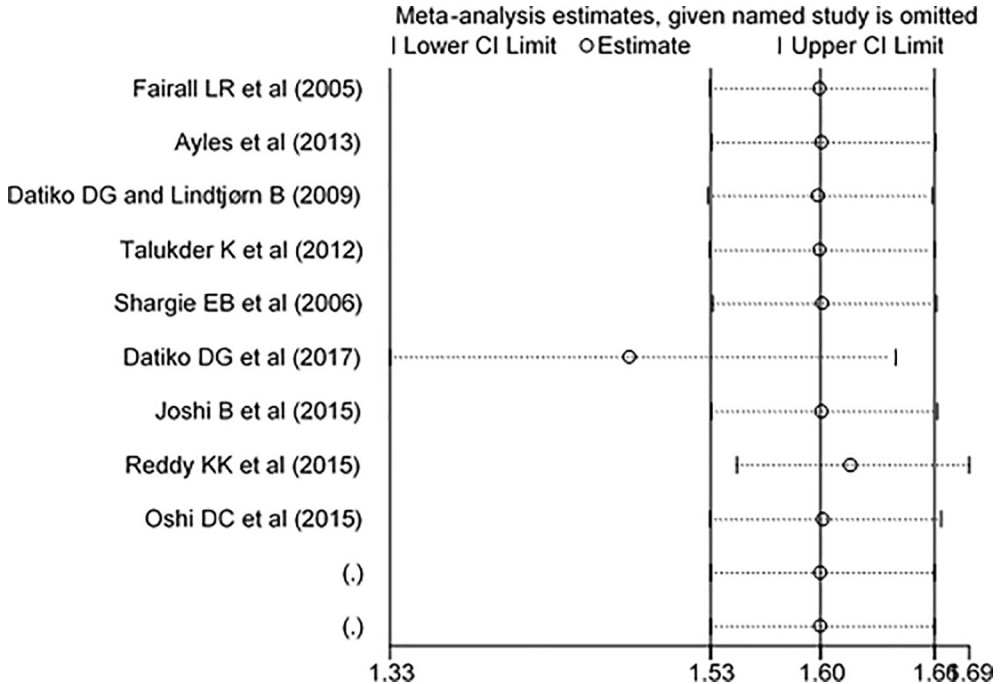

**Fig 3. Sensitivity test on effect of training on tuberculosis case detection.**

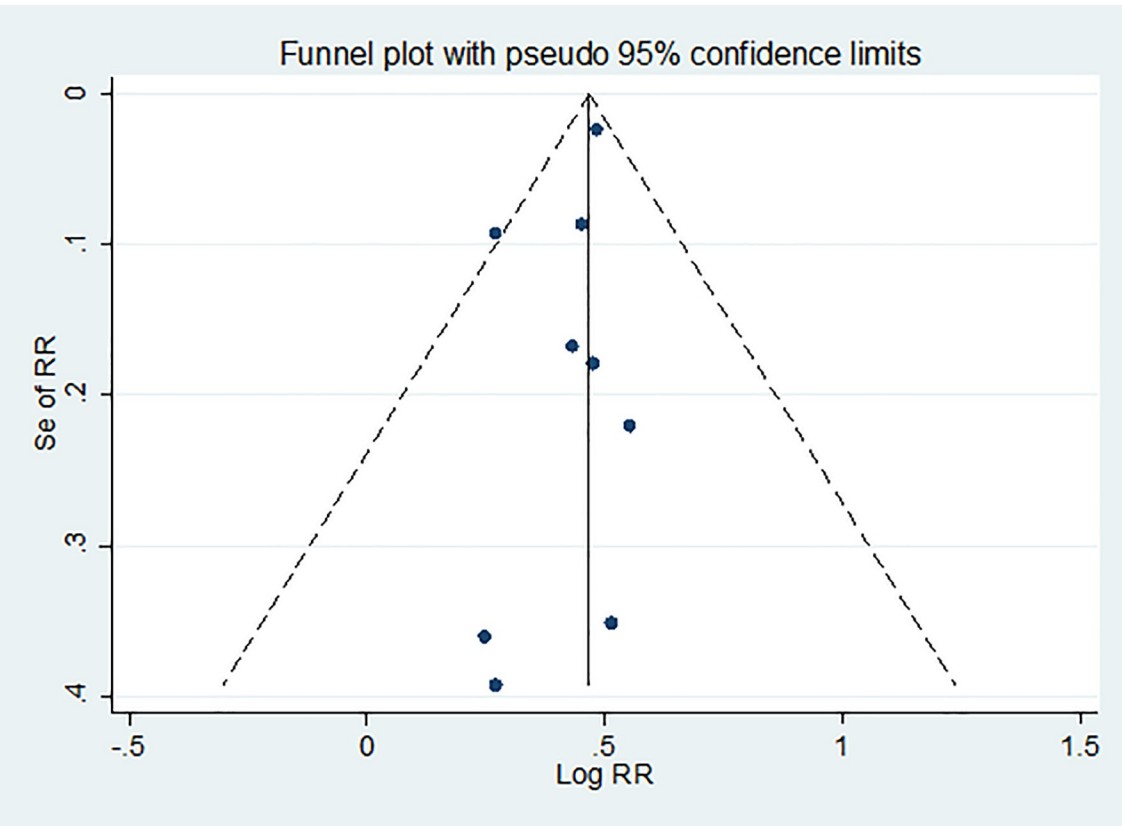

**Fig 4. Funnel plot analysis of publication bias.**

**Table 2. Risk bias assessment using JBI checklist.**

| Authors | Publication year | Randomization Sequence | Baseline characteristics | participants blind to treatment assignment | Blinds of the therapist to treatment | Blinds outcomes assessment | treatment groups treated identically other than the intervention of interest | Completeness of follow up | participants analyzed in the groups to which they were randomized | outcomes measured in the same way for treatment groups | outcomes measured in a reliable way | appropriate statistical analysis | appropriate trial design | Total score |
|---|---|---|---|---|---|---|---|---|---|---|---|---|---|---|
| Fairall L et al. [27] | 2010 | Yes | Yes | No | No | No | Yes | Yes | Yes | Yes | Yes | Yes | Yes | 10 |
| Ayles H et al. [28] | 2013 | Yes | Yes | Yes | Yes | Yes | Yes | Yes | Yes | Yes | Yes | Yes | Yes | 13 |
| Datiko DG et al. [18] | 2009 | Yes | Yes | No | No | No | Yes | Yes | Yes | Yes | Yes | Yes | Yes | 10 |
| Datiko DG et al. [29] | 2017 | Yes | Yes | No | No | No | Yes | Yes | Yes | Yes | Yes | Yes | Yes | 10 |
| Joshi B et al. [33] | 2015 | Yes | Yes | No | No | No | Yes | Yes | Yes | Yes | Yes | Yes | Yes | 10 |
| Talukder K et al [32]. | 2012 | No | No | No | No | No | Yes | Yes | Yes | Yes | Yes | Yes | Yes | 8 |
| Shargie EB et al [30]. | 2006 | Yes | Yes | No | No | No | Yes | Yes | Yes | Yes | Yes | Yes | Yes | 10 |
| Reddy KK et al [19]. | 2016 | Yes | Yes | No | No | No | Yes | Yes | Yes | Yes | Yes | Yes | Yes | 10 |
| Oshi DC et al [31]. | 2016 | No | No | No | No | No | Yes | Yes | Yes | Yes | Yes | Yes | Yes | 8 |

## Discussion

This is the first systematic review and meta-analysis that quantified the effectiveness of providing training for healthcare workers and volunteers on TB case detection. Our findings showed that providing training to healthcare workers and community volunteers significantly increased TB case detection. This is an important finding suggesting the need to scaling-up training to improve case detection and to achieve the global End TB targets [25,34,35].

Community healthcare workers being more familiar with the community and trusted by the community were involved in the intervention. Providing training to community volunteers may improve their knowledge of TB and they can refer suspected cases to TB programs. A previous study found that community-based case detection approach helps to empower patients to deal with their problems [36]. Providing training to health professionals may also be useful for the proper diagnosis and treatment of TB patients [31,32]. Although different training approaches were used their effectiveness in increasing TB case detection were not significantly differ [18,29,31–33].

The World Health Organization (WHO) has a target to end TB by 2035 [37]. The findings from this systematic review provide important evidence that may contribute to achieve this ambitious target. This systematic review revealed that providing effective training to health professionals and community workers can improve TB case detection. It has therefore had a great impact on TB case findings by implementing educational training for healthcare providers and community volunteers. This systematic review and meta-analysis have also implications for health staff and researchers by increasing their knowledge on the effectiveness of training intervention on TB cases detection. Although many people with TB infection remain undiagnosed and had a lack of care [38], our study showed that the implementation of training packages considerably effective approach to case detection and helps to reduce unnecessary delays in diagnosis and received care. The implementation of interventions and using accurate diagnostic tests may improve TB case notification. Therefore, the findings of this study could in turn be implemented into practice.

Trained health workers and community volunteers were the keys to detecting TB cases in the community and primary health care settings and trained community volunteers were crucial to conduct household contact screening in detecting cases and refer cases to nearby TB diagnostic centers [33]. The finding of this systematic review supported the importance of providing educational training to improve the TB case detection rate.

Our review had several limitations. We have included articles published in English only. Most studies included in our study used a sputum smear microscope as a main diagnostic method for TB that mainly identifies positive pulmonary TB, and this type of diagnostic approach has limited ability to identify true positive cases. This evidence is substantiated with another report that conventional tests like microscopic examination have low accurate TB diagnosis as compared to Xpert [39]. Possibly, many people with TB infection are still undetected in resource-limited and TB high burden countries [40]. Study related characteristics were not explored using a meta-regression due to low number of available studies. Some studies show that the training method used is not clear, so it is difficult to know which training method is more effective to increase case detection. We find this to be one of the limitations of the study.

## Conclusions

This systematic review and meta-analysis found that the TB case detection rate can be improved by providing training for healthcare workers and volunteers. Therefore, strengthened educational training interventions should be encouraged to improve TB case detections and to ultimately achieve the End-TB targets.

## Supporting information

**S1 File. PubMed advanced search terms.**
(DOCX)

**S1 Table. Search key terms.**
(DOCX)

**S2 Table. Data collection and risk assessment for RCTs and non-RCTs studies.**
(DOCX)

## Author Contributions

**Conceptualization:** Desalegne Amare, Fentie Ambaw Getahun, Kefyalew Addis Alene.

**Data curation:** Desalegne Amare, Endalkachew Worku Mengesha, Getenet Dessie, Melashu Balew Shiferaw, Kefyalew Addis Alene.

**Formal analysis:** Desalegne Amare, Endalkachew Worku Mengesha, Getenet Dessie, Kefyalew Addis Alene.

**Funding acquisition:** Desalegne Amare.

**Investigation:** Desalegne Amare, Getenet Dessie, Kefyalew Addis Alene.

**Methodology:** Desalegne Amare, Kefyalew Addis Alene.

**Project administration:** Desalegne Amare.

**Resources:** Desalegne Amare.

**Software:** Desalegne Amare.

**Supervision:** Desalegne Amare, Fentie Ambaw Getahun, Kefyalew Addis Alene.

**Validation:** Desalegne Amare.

**Visualization:** Desalegne Amare, Kefyalew Addis Alene.

**Writing – original draft:** Desalegne Amare, Kefyalew Addis Alene.

**Writing – review & editing:** Desalegne Amare, Fentie Ambaw Getahun, Tegenaw Asemamaw Dires, Kefyalew Addis Alene.

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
