## [Decision Letter · Decision Letter 0]

31 Jan 2022

PONE-D-21-37324Effectiveness of healthcare workers and volunteers training on improving tuberculosis case detection: a systematic review and meta-analysisPLOS ONE

Dear Dr. Amare,

Thank you for submitting your manuscript to PLOS ONE. After careful consideration, we feel that it has merit but does not fully meet PLOS ONE’s publication criteria as it currently stands. Therefore, we invite you to submit a revised version of the manuscript that addresses the points raised during the review process.

We look forward to receiving your revised manuscript.

Kind regards,

Paavani Atluri

Academic Editor

PLOS ONE

Journal Requirements:

2. Please confirm that you have included all items recommended in the PRISMA checklist including:

- details of reasons for study exclusions in the PRISMA flowchart and number of studies excluded for each reason. 

(Fentie Ambaw Getahun receives support from AMARI (African Mental Health Research Initiative), which is funded through the DELTAS Africa initiative (DEL-15-01). The DELTAS Africa initiative is an independent funding scheme of the African Academia of Sciences (AAS)’s Alliance for accelerating excellence in Science in Africa (AESA) and supported by the New Partnership for Africa’s Development Planning and Coordinating Agency (NEPAD Agency) with funding from the Wellcome Trust (DEL-15-01) and the UK government. The views expressed in this publication are those of the authors and do not necessarily those of the AAS, NEPAD Agency, Wellcome Trust or the UK government.)ment that declares *all* the funding or sources of support (whether external or internal to your organization) received during this study, as detailed online in our guide for authors at http://journals.plos.org/plosone/s/submit-now.  Please also include the statement “There was no additional external funding received for this study.”)

Please provide an amended state in your updated Funding Statement. 

(Fentie Ambaw Getahun receives support from AMARI (African Mental Health Research Initiative), which is funded through the DELTAS Africa initiative (DEL-15-01). The DELTAS Africa initiative is an independent funding scheme of the African Academia of Sciences (AAS)’s Alliance for accelerating excellence in Science in Africa (AESA) and supported by the New Partnership for Africa’s Development Planning and Coordinating Agency (NEPAD Agency) with funding from the Wellcome Trust (DEL-15-01) and the UK government. The views expressed in this publication are those of the authors and do not necessarily those of the AAS, NEPAD Agency, Wellcome Trust or the UK government.)

(The funders had no role in study design, data collection and analysis, decision to publish, or preparation of the manuscript.)

6. Please amend either the abstract on the online submission form (via Edit Submission) or the abstract in the manuscript so that they are identical.

7.We note that this manuscript is a systematic review or meta-analysis; our author guidelines therefore require that you use PRISMA guidance to help improve reporting quality of this type of study. Please upload copies of the completed PRISMA checklist as Supporting Information with a file name “PRISMA checklist”.

Reviewers' comments:

Reviewer's Responses to Questions

**Comments to the Author**

1. Is the manuscript technically sound, and do the data support the conclusions?

Reviewer #1: Partly

Reviewer #2: Yes

2. Has the statistical analysis been performed appropriately and rigorously? 

Reviewer #1: Yes

Reviewer #2: Yes

3. Have the authors made all data underlying the findings in their manuscript fully available?

Reviewer #1: Yes

Reviewer #2: Yes

4. Is the manuscript presented in an intelligible fashion and written in standard English?

Reviewer #1: No

Reviewer #2: Yes

5. Review Comments to the Author

Reviewer #1: This is an interesting review of several studies showing that training HCWs leads to increased TB case detection. The hypothesis and conclusion are expected, therefore this analysis is not surprising or unexpected in its results.

This reviewer would like to see a more in-depth analysis of the studies, rather than the superficial one where the conclusions state more training helps increase TB case detection. Based on the strength of the studies included, did you see certain training methods be more successful than others? What were the training methods used? The results sections seems a little abrupt in its presentation.

The article also needs to be proof-read. There are grammatical and spelling errors in places, some of which I have listed below:

Line 25 - Is should be unbold

Line 27 - professionals

Line 29 - suggest removing the term RCT as your inclusion criteria also included trials which were not RCTs

Line 56 - second only to COVID19 may not remain true in the long run

Line 58 - remove "were"

Line 61 - replace "to" with of

Line 66 - diagnosis

Line 75 - replace "are investigating" with have investigated

Lines 131-134 - there are grammatical errors here

Line 171 - training

Discussion:

Line 5- workers

Reviewer #2: Very good topic as a lot of health care professionals and volunteers do not focus on TB due to fewer number of cases in the USA when compared to other countries like India.

Statistical analysis was appropriate and presented well.

6. PLOS authors have the option to publish the peer review history of their article (what does this mean?). If published, this will include your full peer review and any attached files.

Reviewer #1: **Yes: **Pranjali Sharma, MD

Reviewer #2: **Yes: **Praveena Jaidev

---

## [Author Response · Author response to Decision Letter 0]

15 Feb 2022

Rebuttal letter

We thank the editor and the two reviewers for their comments on our manuscript. The table below is our response to each point raised by the academic editor and reviewers. We hope that we satisfyingly addressed them and that the manuscript will be now suited for publication.

Sincerely,

Desalegne Amare

 

S.No Questions Responses 

 Editor’s recommendation 

1 Ensure that your manuscript meets PLOS ONE's style requirements, including those for file naming We have arranged documents according to the PLOS ONE’s style requirement. The name of the files are also arranged based on PLOS ONE’s style for example supplementary files names are corrected as S1 File, S1 Table, S2 Table, Fig 1, Fig2, Fig3 & Fig4

2 Please confirm that you have included all items recommended in the PRISMA checklist PRISMA checklist was used to include all items. And also we followed PRISMA flow chart. 

3 Please provide an amended state in your updated Funding Statement According to your recommendation, funding statement is amended as “Fentie Ambaw Getahun receives support from AMARI (African Mental Health Research Initiative), which is funded through the DELTAS Africa initiative (DEL-15-01). The DELTAS Africa initiative is an independent funding scheme of the African Academia of Sciences (AAS)’s Alliance for accelerating excellence in Science in Africa (AESA) and supported by the New Partnership for Africa’s Development Planning and Coordinating Agency (NEPAD Agency) with funding from the Wellcome Trust (DEL-15-01) and the UK government. The views expressed in this publication are those of the authors and do not necessarily those of the AAS, NEPAD Agency, Wellcome Trust or the UK government.)”

4 Funding information should not appear in the Acknowledgments section The statement that described under acknowledges section is removed because it was misplaced. So the statement is moved to funding session. 

5 We note that you have stated that you will provide repository information for your data at acceptance. Should your manuscript be accepted for publication, we will hold it until you provide the relevant accession numbers or DOIs necessary to access your data. If you wish to make changes to your Data Availability statement, please describe these changes in your cover letter There is no published data repository elsewhere. “All data is available in the manuscript”. 

6 Please amend either the abstract on the online submission form (via Edit Submission) or the abstract in the manuscript so that they are identical We made identical/the same abstract on the online submission form and in the manuscript. 

7 Please upload copies of the completed PRISMA checklist as Supporting Information with a file name “PRISMA checklist We have done that PRISMA checklist as Supporting Information with a file name “PRISMA checklist”.

 

Reviewer #1

8 Did you see certain training methods be more successful than others? In the review process we faced difficulty to determine which training methods are more effective than the other because some of the studies do not show a clear training methods. 

9 What were the training methods used? • Fairall LR 2005: Educational training at health centres and community meetings with messages on childhood TB, using different teaching materials a child TB flip chart with pictures and text messages in the local language. 

• Talukder 2012: Demonstration on Keith Edwards Child TB score chart, administration of the Mantoux test 

• Daniel G. Datiko 2009, 2007: On job training and health education sessions. 

• Reddy 2015: Trained community volunteers but the method of training is not described clearly.

• Oshi DC 2016: Training was provided to improve the knowledge and skill of health care workers in identifying presumptive childhood TB, but method of training is not clear. 

• Shargie 2006: Group based interactive lecturing and demonstration and the training focused on;

basic facts about TB: its cause, transmission, symptoms, diagnosis, prevention, treatment, and outcomes

societal perceptions about TB, on how to identify and refer a symptomatic TB suspect, and on how to communicate basic and locally understandable messages about TB 

Case-finding, diagnostic procedures, outreach coordination, handling of sputum specimens, interview techniques, and record-keeping.

• Ayles 2013: The method of training is not clear 

10 Line 25 - Is should be unbold On line 25 the verb “is” is unbold 

11 Line 27 – professionals The word professional changed to “professionals” 

12 Line 29 - suggest removing the term RCT as your inclusion criteria also included trials which were not RCTs Both RCT and Non-RCT are added 

13 Line 56 - second only to COVID19 may not remain true in the long run Despite Tuberculosis (TB) is a preventable and treatable disease, it remains an important cause of death from an infectious agent, especially, after COVID-19, it is the second deadliest infectious killer. 

14 Line 58 - remove "were" “Were” is removed 

15 Line 61 - replace "to" with of Done 

16 Line 66 – diagnosis Diagnoses changed diagnosis 

17 Line 75 - replace "are investigating" with have investigated Are investigating changed with “have investigated”

18 Lines 131-134 - there are grammatical errors here The authors considered low risk of bias ‘Yes’ either the study protocol is available and all of the studies’ pre-specified outcomes that are of interest in the review have been reported in the pre-specified way or the study protocol is not available but it is clear that the published reports include all of the study’s pre-specified outcomes and all expected outcomes that are of interest in the review. Authors considered high risk of bias ‘No’ if any of the following is fulfilled: Not all of the study’s pre-specified primary outcomes have been reported, one or more primary outcomes is reported using measurements analysis methods or subsets of the data that were not pre-specified, one or more reported primary outcomes were not pre-specified, one or more outcomes of interest in the review are reported incompletely, the study report fails to include results for a key outcome that would be expected to have been reported for such a study. Authors considered unclear if there is insufficient information to permit judgment of ‘Yes’ or ‘No’

19 Line 171 – training “Trainings” changed to “training” 

20 Discussion:

Line 5- workers The word “works” changed “workers”

Reviewer 2

21 Very good topic as a lot of health care professionals and volunteers do not focus on TB due to fewer number of cases in the USA when compared to other countries like India No comments 

22 Statistical analysis was appropriate and presented well. No comments

---

## [Decision Letter · Decision Letter 1]

15 Jun 2022

PONE-D-21-37324R1Effectiveness of healthcare workers and volunteers training on improving tuberculosis case detection: a systematic review and meta-analysisPLOS ONE

Dear Dr. Amare,

Thank you for submitting your manuscript to PLOS ONE. After careful consideration, we feel that it has merit but does not fully meet PLOS ONE’s publication criteria as it currently stands. Therefore, we invite you to submit a revised version of the manuscript that addresses the points raised during the review process. Please submit your revised manuscript by Jul 30 2022 11:59PM. If you will need more time than this to complete your revisions, please reply to this message or contact the journal office at plosone@plos.org. Please include the following items when submitting your revised manuscript:A rebuttal letter that responds to each point raised by the academic editor and reviewer(s). You should upload this letter as a separate file labeled 'Response to Reviewers'.A marked-up copy of your manuscript that highlights changes made to the original version. You should upload this as a separate file labeled 'Revised Manuscript with Track Changes'.An unmarked version of your revised paper without tracked changes. You should upload this as a separate file labeled 'Manuscript'.If applicable, we recommend that you deposit your laboratory protocols in protocols.io to enhance the reproducibility of your results. Protocols.io assigns your protocol its own identifier (DOI) so that it can be cited independently in the future. For instructions see: https://journals.plos.org/plosone/s/submission-guidelines#loc-laboratory-protocols. Additionally, PLOS ONE offers an option for publishing peer-reviewed Lab Protocol articles, which describe protocols hosted on protocols.io. Read more information on sharing protocols at https://plos.org/protocols?utm_medium=editorial-email&utm_source=authorletters&utm_campaign=protocols.

We look forward to receiving your revised manuscript.

Kind regards,

Paavani Atluri

Academic Editor

PLOS ONE

Journal Requirements:

Reviewers' comments:

Reviewer's Responses to Questions

**Comments to the Author**

1. If the authors have adequately addressed your comments raised in a previous round of review and you feel that this manuscript is now acceptable for publication, you may indicate that here to bypass the “Comments to the Author” section, enter your conflict of interest statement in the “Confidential to Editor” section, and submit your "Accept" recommendation.

Reviewer #1: All comments have been addressed

Reviewer #3: (No Response)

2. Is the manuscript technically sound, and do the data support the conclusions?

Reviewer #1: Yes

Reviewer #3: Partly

3. Has the statistical analysis been performed appropriately and rigorously? 

Reviewer #1: Yes

Reviewer #3: No

4. Have the authors made all data underlying the findings in their manuscript fully available?

Reviewer #1: No

Reviewer #3: Yes

5. Is the manuscript presented in an intelligible fashion and written in standard English?

Reviewer #1: Yes

Reviewer #3: No

6. Review Comments to the Author

Reviewer #1: All previously raised comments have been addressed. The data requested should be made available when the manuscript is accepted. But otherwise, no additional revisions recommended.

Reviewer #3: Present study titled ‘Effectiveness of healthcare workers and volunteers training on improving tuberculosis case detection: a systematic review and meta-analysis’ is an interesting study focusing on training among health care workers. However, proofreading for English, grammatical and punctuations error are recommended. Also, there are serious discrepancies in the referencing of articles throughout the manuscript. I have highlighted some of those below:

1. Abbreviation and expansion should be mentioned at their first appearance such as WHO.

2. Grammatical and punctuation errors such as ‘,’ in page 3 line 60; page 5 line 119.

3. Repetitive authors list in the bracket page 4 line 116

4. Ref 2 and 3 is same. Please correct the bibliography accordingly.

5. Please update the references no 3,4 and 7. Please provide an updated estimates as WHO TB report 2021 is available.

6. Ref 4 and 5 is inaccurate. Correct citation would be ‘Daniel P Chin, Christy L Hanson, Finding the Missing Tuberculosis Patients, The Journal of Infectious Diseases, Volume 216, Issue suppl_7, 1 October 2017, Pages S675–S678’

7. In page 3 line 60, 40 % of these cases is not clear. If authors are referring to ‘incidence’ or ‘prevalence’ cases. Please correct it.

8. Ref 4 and 14 is same. Also, citation seems not appropriate for ref 14 in introduction and discussion section.

9. Page 4, line 105, ref 26 is not correct.

10. List of abbreviation should be complete such TB is missing in the list

11. Fig 1 is not accurate. Screening section should be corrected. Exclusion shell (n =1946) should be after articles screened (n = 2015).

12. Please suggest, Ref 32 Ayles et al (2013) has reported prevalence of the TB cases, not the TB incidence. Why this has been included in the selected studies. Primary outcome of interest of meta-analysis is proportion of new cases (page 5 line 126-127).

13. Please suggest, why RCT and Non-RCT studies were meta-analysed together. This would have introduced measuring bias in the effect estimates.

Overall, this manuscript required a major revision especially in terms of referencing the correct citations.

7. PLOS authors have the option to publish the peer review history of their article (what does this mean?). If published, this will include your full peer review and any attached files.

Reviewer #1: **Yes: **Pranjali P Sharma

Reviewer #3: **Yes: **Prabal Chourasia

---

## [Author Response · Author response to Decision Letter 1]

21 Jun 2022

Jun 20, 2022

Editor 

PloS ONE 

Subject: Submission of revised manuscript

Thank you for the opportunity to revise our manuscript” Effectiveness of healthcare workers and volunteers training on improving tuberculosis case detection: a systematic review and meta-analysis”. We are grateful for the reviewers’ and editor's comments and believe that the revised manuscript is stronger as a result of their feedback. We are pleased to see that our study is well-recognized by the reviewers as a significant contribution. Below, we have provided point-by-point responses including how and where the text was modified in the main manuscript. Two separate documents, one with track change and the other a clean version are uploaded to the system. We hope that the revised version is now suitable for publication and look forward to hearing from you in due course. 

Desalegne Amare 

Reviewer #1: All previously raised comments have been addressed. The data requested should be made available when the manuscript is accepted. But otherwise, no additional revisions are recommended.

Response: all relevant documents including the data, quality assessment tools, and risk of bias assessment tools are now included in the main document and supplementary files. 

Reviewer #3: Present study titled ‘Effectiveness of healthcare workers and volunteers training on improving tuberculosis case detection: a systematic review and meta-analysis is an interesting study focusing on training among healthcare workers. However, 

1. Abbreviations and expansion should be mentioned at their first appearance such as WHO.

Response 1: We have now spelled out all the abbreviations when they appear for the first time in the manuscript. On page 3 line 60, we have now presented the expanded forms of WHO as ‘World Health Organization (WHO)’

2. Grammatical and punctuation errors such as ‘,’ on page 3 line 60; page 5 line 119.

Respones2: We have thoroughly revised the manuscript and edit all the grammatical and punctuation errors. The statements on page 3 lines 60-65 have been now updated as ‘According to the World Health Organization (WHO) report, approximately 10 million people who developed TB in 2020. According to the WHO, 4.1 million people have tuberculosis but have not been diagnosed or have not reported the disease to national authorities. India (41%), Indonesia (14%), the Philippines (12%), and China (8%) were the countries that contributed the most to the global reduction in TB notifications between 2019 and 2020(2)’. 

The stamen on page 5, line 119 has been also now revised as ‘The spreadsheet was checked and refined before data extraction’

3. Repetitive authors list in the bracket on page 4 line 116

 Response3: authors list is now corrected as (MBS, EWM, TAD, and FAG).

4. Refs 2 and 3 are the same. Please correct the bibliography accordingly. 

Response 4: The bibliography is now corrected as suggested.

5. Please update references no 3, 4, and 7. Please provide an updated estimate as the WHO TB report 2021 is available.

Response 5: The references 3, 4, and 7 are now updated according to the WHO report 2021. 

6. Ref 4 and 5 is inaccurate. The correct citation would be ‘Daniel P Chin, Christy L Hanson, Finding the Missing Tuberculosis Patients, The Journal of Infectious Diseases, Volume 216, Issue suppl_7, 1 October 2017, Pages S675–S678’

Response 6: We have also checked and revised the citation to the appropriate statements. The statement “As the global community works to end the TB epidemic, the most pressing challenge is that more than 4 million TB patients each year, 40% of all incident cases are not reported to NTPs” is not mentioned by Daniel P Chin and Christy L Hanson as the original source, but these authors used this statement by acknowledging 2016 WHO report. Similarly, 2021 WHO report stated this “4.1 million people have tuberculosis but have not been diagnosed or have not reported the disease to national authorities”. Thus, we used the original sources of information for our citation. 

7. On page 3 line 60, 40 % of these cases is not clear. If authors are referring to ‘incidence’ or ‘prevalence’ cases. Please correct it.

Response 7: This statement refers to the total number of new TB cases (10 million) worldwide, about 4 million about 40% of the total cases were not reported. Thus, it is not about incidence or prevalence, but it is about the absolute number of new cases missed in the annual report. 

8. Refs 4 and 14 are the same. Also, citation seems not appropriate for ref 14 in the introduction and discussion section.

Response 8: Thank you it is corrected accordingly. 

9. On page 4, line 105, ref 26 is not correct. 

Response 9: Thank you for this comment again. It is now corrected as “Higgins JP, Thomas J, Chandler J, Cumpston M, Li T, Page MJ, et al. Cochrane handbook for systematic reviews of interventions: John Wiley & Sons; 2019.”

10. List of abbreviations should be complete such as TB is missing in the list

Response 10: We have already provided this abbreviation online

11. Fig 1 is not accurate. The screening section should be corrected. Exclusion shell (n =1946) should be after articles screened (n = 2015).

Response 11: Thank you for this comment. The flow chart showing the screening process of the articles in now revised and corrected as suggested by the review. Box contains excluded articles after reviewing by using titles and abstracts is placed under-screened articles (2015). 

12. Please suggest that Ref 32 Ayles et al (2013) has reported the prevalence of the TB cases, not the TB incidence. Why this has been included in the selected studies. The primary outcome of interest in meta-analysis is the proportion of new cases (on page 5 lines 126-127).

Response 12: This study is a randomized control trial that has both intervention and control groups, thus relative risk was calculated manually from the cases and control groups. 

13. Please suggest, why RCT and Non-RCT studies were meta-analyzed together. This would have introduced measuring bias in the effect estimates.

Response 13: This study includes all interventional studies that might be RCT or NRCT that provided important information for our outcome of interest. However, to avoid bias, we have used strong bias assessment tools.

---

## [Editor Report · Decision Letter 2]

8 Jul 2022

Effectiveness of healthcare workers and volunteers training on improving tuberculosis case detection: a systematic review and meta-analysis

PONE-D-21-37324R2

Dear Dr.Amare,

We’re pleased to inform you that your manuscript has been judged scientifically suitable for publication and will be formally accepted for publication once it meets all outstanding technical requirements.

Kind regards,

Paavani Atluri

Academic Editor

PLOS ONE
---

## [Editor Report · Acceptance letter]

2 Sep 2022

PONE-D-21-37324R2 

Effectiveness of healthcare workers and volunteers training on improving tuberculosis case detection: a systematic review and meta-analysis 

Dear Dr. Amare:

I'm pleased to inform you that your manuscript has been deemed suitable for publication in PLOS ONE. Congratulations! Your manuscript is now with our production department. 

Kind regards, 

on behalf of

Dr. Paavani Atluri 

Academic Editor

PLOS ONE